# Structural Achievability of an NH–π Interaction between Gln and Phe in a Crystal Structure of a Collagen-like Peptide

**DOI:** 10.3390/biom12101433

**Published:** 2022-10-06

**Authors:** Ruixue Zhang, You Xu, Jun Lan, Shilong Fan, Jing Huang, Fei Xu

**Affiliations:** 1Ministry of Education Key Laboratory of Industrial Biotechnology, School of Biotechnology, Jiangnan University, Wuxi 214122, China; 2Key Laboratory of Structural Biology of Zhejiang Province, School of Life Sciences, Westlake University, Hangzhou 310024, China; 3Westlake AI Therapeutics Lab, Westlake Laboratory of Life Sciences and Biomedicine, Hangzhou 310024, China; 4Center for Structural Biology, School of Life Sciences, Tsinghua University, Beijing 100084, China

**Keywords:** NH–π interactions, side chain interactions, X-ray crystallography, collagen, molecular dynamics, quantum chemistry, circular dichroism, structural bioinformatics

## Abstract

NH–π interactions between polar and aromatic residues are well distributed in proteins whose stabilizing effects have been investigated in globular and fibrous proteins. In order to gain structural insights into side chain NH–π interactions, we solved a crystal structure of a collagen-like peptide containing Gln-Phe pairs. The Gln-Phe NH–π interactions were further characterized by quantum calculations, molecular simulations, and structural bioinformatics. The analyses indicated that the NH–π interactions are robust under various solvent conditions, can be distributed either on the protein surface or in its hydrophobic core and can form at a wide range of distances between residues. This study suggested that NH–π interactions can play a versatile role in protein design, including engineering hydrophobic cores, solvent accessible surfaces, and protein–protein interfaces.

## 1. Introduction

NH groups are prevalent in proteins given their location in both the backbone and some side chains (e.g., Lys, Arg, Asn, and Gln). A previous study showed that approximately 1 in 10.8 aromatic side chains, not including His, make close contact with an NH group [1]. The nature of NH–π interactions is largely dependent on the polarity of the donor group. When N is from ammonium/guanidinium of Lys/Arg, the interaction energy is dominated by an electrostatic term, so-called cation–π interactions. The stabilizing effects of side chain cation–π interactions have been characterized and subsequently used to enhance protein stability [2,3,4,5]. However, when the N from either the backbone or uncharged Asn/Gln side chains is best classified between the conventional hydrogen bonding and a weaker interaction dominated by dispersion. Burley et al. initially noted that aromatic residues could make close contact with both Arg/Lys or Asn/Gln with similar preference [6], suggesting that side chain NH–π interactions could also play an important role in protein stability and/or function. The quantification of NH–π interactions in small molecule systems had been determined experimentally, which was also supported by quantum mechanical theory. In NH–π interactions in the N-methylformamide-benzene complex, the minimum energy distance was 3.2 Å and the binding energy was −4.37 kcal/mol [7]. In contrast, the stabilizing effects of NH–π interactions in Acetyl-Phe-Gly-Gly-N-Methyl amide peptide model ranged from −1.98 to −9.24 kcal/mol [8]. More detailed studies are needed to understand the NH–π interactions in proteins.

Collagen-like peptides have been utilized as a molecular platform to investigate side chain interactions. The characteristic structure of a collagen-like peptide is a triple helix composed of three chains. Each chain adopts a well-extended PPII conformation with successive Gly-X-Y triplets [9,10,11]. Although the X and Y positions of this motif can accommodate any amino acid, the highest stability triplet is Gly-Pro-Hyp, where Hyp (O) represents (4R)-hydroxyproline [12]. Chain trimerization is induced by inter-chain backbone hydrogen bonding between Gly and non-Gly residues [9]. In this context, Gly is buried inside the triple helices, while side chains of residues in the X/Y positions are exposed on the collagen surface. With host–guest methods, various amino acids are introduced to replace Pro at X or Hyp at Y-position at the central triplets of the host sequences containing high-imino triplets only, such as (GPO)_n,_ where n = 8, 10 [13,14,15,16,17]. As GPO is the most stabilizing triplet, any substitution could destabilize the peptides with decreased melting temperature (*T*_m_). When a peptide containing two substitutions exhibits a higher *T*_m_ than expected one by adding the destabilizing effects of two individual substitutions, which could be taken as the energetic contribution of side chain interactions [13]. Stabilizing effects of salt bridges and cation–π interactions, such as Arg-Phe and Lys-Phe pairs, have also been characterized by the host–guest methods [5,14,18,19,20,21].

In collagen triple helices, there are two types of inter-chain side chain interaction between two neighboring chains containing the GXYGX’Y’ sequence. One type is lateral (Y-X) interaction and the other type is axial (Y-X’). Walker et al. designed guest peptides to calculate the energetic contributions of the axial and lateral interactions [22]. The guest sequence GPOGFQ embedded in (GPO)_8_ contained two lateral Gln-Phe pairs with *T*_m_ 7.5°C lower than the host, while GPQGFO contained one lateral and two axial Gln-Phe pairs with *T*_m_ similar to the host [23]. The energetic contribution of the axial and lateral Gln-Phe pair was derived to be 2.13 °C and −0.75 °C, respectively. Axial Gln-Phe pairs were also utilized to induce the formation of a collagen heterotrimer. Molecular dynamics showed that axial Gln-Phe pairs could make close contacts (3.8 Å). However, more detailed structural information is necessary to understand the properties of the NH–π interactions in collagen.

In this study, in order to gain structural insights into an NH–π interaction between Gln and Phe and better understand its chemical properties, we solved a crystal structure of a collagen-like peptides (GPO)_3_GPQGFO(GPO)_3_, abbreviated as GPQGFO, and characterized the side chain interactions using several lines of evidence including circular dichroism, quantum chemical analysis, molecular dynamics simulation, and structural bioinformatics. Two major features of the NH–π interaction were revealed, including its environmental stability and conformational diversity, which provide a framework for exploring the mechanisms of non-covalent XH–π (X = N, S, O, C) interactions and further applying them to protein design.

## 2. Materials and Methods

### 2.1. Peptide Synthesis and Purification

The peptides were synthesized with FMOC solid-phase chemistry method by GL Biochem Ltd (Shanghai, China), and purified to more than 95% purity. The purity was tested with reversed-phase high-performance liquid chromatography (RP-HPLC) on an Agilent 1260 system using an AdvanceBio peptide map column (C18), 2.1 × 150 mm, pore size 2.7-micron, with a linear gradient of water and acetonitrile gradient each containing 0.5% TFA. Molecular weights were validated with matrix-assisted laser desorption ionization time-of-flight mass spectrometry (MALDI-TOF MS) on a Bruker autoflex II using a prespotted anchor chip with 10 mg/ml 2,5-dihydroxybenzoic acid in water, 50% acetonitrile as the matrix, and a dual-stage reflection electron microscope obtain the spectrum (Appendix A). The sequence of the peptide was as below:

GPQGFO: GPOGPOGPOGP**Q**G**F**OGPOGPOGPO

where O is (4R)-hydroxyproline. The N- and C- termini are acetylated and amidated, respectively.

### 2.2. Crystallization and Data Collection

The pure and lyophilized GPQGFO peptide powder was dissolved in a 20 μL buffer at pH 7.0 with 10 mM Tris·HCl making the solutions of 5 mM concentration. Incubate at 4 °C overnight and then set for crystallization by using the hanging drop vapor diffusion method at 4 °C. After ~one month, the best crystals were obtained under 0.2 M potassium nitrate and 20% PEG 3350. X-ray data were collected at 100 K, and the best one was diffracted to 1.385 Å. The space group was P2_1_22_1_, with dimensions: a = 24.09 Å, b = 28.01 Å, c = 73.37 Å, α = β = γ = 90° indexed by the HKL2000 software (HKL Research, Charlottesville, VA, USA). Data collection and processing statistics are summarized in Appendix A.

### 2.3. Structure Determination and Refinement

The structures of GPQGFO were solved by molecular replacement using a Phaser from the Phenix software suite [24] with a collagen structure (PDB ID: 5YAN [25]) as the search models. Initial phases were improved by rigid body refinement followed by rounds of simulated annealing and anisotropic B-factor refinement using the Phenix suite. Manual model rebuilding was performed using COOT [26]. Further rounds of refinement were performed using the phenix.refine program implemented in the PHENIX package with coordinate refinement, isotropic ADP refinement, and bulk solvent modeling. The final GPQGFO model contains 598 peptide atoms and 115 water molecules. The final GPQGFO structure has an R_work_/R_free_ value of 16.7/18.0%.

### 2.4. Circular Dichroism (CD) Spectra

CD was recorded in 1-mm quartz cuvettes on a Chirascan instrument with a six-cell thermostated cell holder and a temperature controller (Applied Photophysics Ltd, London UK). Peptide solutions of 0.2 mM were characterized in phosphate-buffer (at pH 7.0) or in pH 3, 9 or in 100, 150, 300, 500 mM NaCl. Wavelength scans were conducted from 190 to 260 nm at 4 °C. For temperature-induced denaturation, ellipticity was measured from 4 to 80 °C with a heating rate of 1 °C for 6 min. Observed ellipticity was converted to molar ellipticity by dividing raw values by the peptide concentration, number of residues, and cell path length. The melting temperature, *T*_m_, was estimated as below:(1)$$\begin{array}{c} F\left(T\right)=\frac{\theta \left(T\right)-\theta_{U}\left(T\right)}{\theta_{F}\left(T\right)-\theta_{U}\left(T\right)} \end{array}$$
where *θ(T)* is the observed ellipticity, and *θ_F_(T)* and *θ_U_(T)* are predicted ellipticities derived from linear fits to the folded and unfolded baselines. The melting temperature is estimated as the T where F(T) = 0.5.

### 2.5. Quantum Mechanical (QM) Analysis

To probe the global energy landscape of Q-F interaction, a three-dimensional potential energy scan was performed between acetamide and toluene. A set of 3-D Cartesian axes was created where the x-y plane was mapped on the toluene ring with atom C1 defined as the origin and vector CM-C1 as the *x*-axis. Because toluene is symmetric about the *x*-axis and also the x-y plane, the whole interaction space between two molecules can be represented by such a quadrant that on only one side of the *x*-axis and x-y plane. The space was gridded at every 0.4 Å from 0 to 4.8 Å along x- and y-axes and every 0.2 Å from 2.0 Å to 5.0 Å along the *z*-axis. In each grid, the coordinate of the N atom in acetamide was constrained while the other atoms were free to move. In total, 2704 pairs of molecule models were optimized and calculated and the energy surface of the amide position related to the toluene ring was scanned. The M06-2X functional with cc-pVTZ basis set was employed in geometric optimization using Gaussian. Then the single point energy was calculated with the aug-cc-pVTZ basis set and counterpoise correction was employed to calculate the interaction energy. ORCA [27] was used to perform all single-point energy calculations.

Additional QM calculations were carried out to understand the NH–π interaction between acetamide and toluene. Geometry optimization was carried out at the MP2/aug-cc-pVTZ level using psi4 [28] and interaction energy was obtained at the DLNPO-CCSD(T)/aug-cc-pVQZ level with counterpoise correction using ORCA. SAPT calculations were carried out with the aug-cc-pVQZ basis set using psi4.

### 2.6. Calculation of Geometric Parameters

To determine the geometric relationship between the pairwise, three bond conformations (*d*_NM_, θ, ω,) and three side-chain-positions (*d*_Cγ-π_, τ, χ) were defined. *d*_NM_ was denoted as the distance between the N of the amide NH groups of Gln or acetamide and the center (M) of the phenyl ring. At this time, the angle between NH and M was defined as ω (0 < ω < 180°), and the acute angle between the MN vector and the normal vector of the phenyl ring was described as θ (0 < θ < 90°). The *d*_Cγ-π_ and τ were defined as the shortest distance and angle between Cγ of the Gln or acetamide and the plane formed by the phenyl ring. If Cγ is located in the same direction as the normal vector, τ is positive (0 < τ < 90°), otherwise, it is negative (–90° < τ < 0°). Whereas the χ was described as the angle between the vector formed by the projection of the Cγ-N vector onto the plane and the Cβ-Cγ vector formed by the phenyl ring. See SI method for calculation details. All atomic cartesian coordinates were retrieved from the pdb files. Since hydrogen atoms are missing from most pdb files, theoretical positions of hydrogen atoms (X-H = 1.0 Å) were then added with HGEN [29] software, but only those at positions allowing this calculation unambiguously were used further. The following cutoffs are used to screen the Q-F pairs with the NH–π interactions, *d*_N-M_ < 4.3 Å, θ < 25°, ω > 120° [1].

### 2.7. Molecular Dynamics (MD) Simulation

The molecular triplex model with 24 residues in each strand was built from the crystal structure using CHARMM [30]. After energy minimization in a vacuum, the system was solvated with TIP3P water [31] in a cubic box that was 89 Å in length. No counterions or salt concentration was included. Periodic boundary conditions were used and the particle mesh Ewald was applied on both electrostatic [32] and Lennard–Jones (LJ-PME) [33,34] interactions, with 12 Å as the real space cutoff. Applying CHARMM36m [35,36] force field, the simulation was performed in an isothermal-isobaric ensemble (NPT) at 298 K and 1 atm by employing the Andersen thermostat and Monte Carlo barostat, respectively. With the constraints on hydrogen-involved bonds, a time step of 2 fs was used in the velocity Verlet integrator. After 0.64 ns equilibrium simulation, the system was further run for 200 ns using OpenMM [37] with snapshots saved every 100 ps.

### 2.8. Database Analysis of Q-F Pair Interaction

A non-redundant set was curated containing 9658 protein structures all determined by X-ray diffraction to a resolution of 2.0 Å or higher, refined to R ≤ 0.25 and less than 30% sequence homology by PIESCES suits [38,39].

The following criteria were employed to assemble the set: (1) no theoretical model structures and no incomplete entries, i.e., Cα coordinates only, were accepted; (2) only crystal structures with a resolution of 2.0 Å or better and a crystallographic R-factor of 0.25 or lower were accepted; (3) the minimum chain length was 40 amino acid residues; and (4) the maximum acceptable amino acid identity between any two protein chains of the set was 30%.

## 3. Results

### 3.1. Triple Helices Remained Stable under Various Ionic Strengths and pH

In a previous study of GPQGFO peptides, it was indicated that a favorable Q–F interaction could be generated to stabilize the triple helix in a neutral environment [23]. To experimentally assess the tolerance of the Q-F pairs in different environments, the thermal stability of GPQGFO under various NaCl (100–500 mM) concentrations and pH (3, 7, 9) was characterized by circular dichroism spectra. The molar residual ellipticity (MRE) at 220 nm and *T*_m_ (~40°C) were apparently unaffected (Figure 1b,c). These results indicated that the stability contribution of the Q-F pairs was maintained under various environmental conditions, which distinguished the GPQGFO motif from that of less tolerant motifs such as GPKGEO where the stabilizing electrostatic interactions are more affected [18]. Notably, the *T*_m_ values of peptides recorded here were higher than those reported by Persikov et al. [12] and Walker et al. [23], potentially due to differences in melting schedule, host sequence, or blocking of peptide termini.

### 3.2. Side Chain Conformations in Crystal Structure

In order to gain a better understanding of the Q-F pairs in the context of a triple helix, and especially to help understand the difference between the axial and lateral sequential geometry [23]. The GPQGFO peptide crystallizes with the triple helical structure characteristic of collagen molecules with 1.385 Å resolution. More data acquisition and refinement statistics are available in Appendix A. Two axial pairs and one lateral were observed in the structure depending on the relative positions of the interacting amino acids (Figure 1e). Two axial pairs were, respectively, composed of Gln in the leading and Phe in the middle chains of the triple helices, the other was between the middle and trailing chains. Three bond-conformation parameters, *d*_NM_, θ, and ω, describe whether the Gln Nε-H perpendicularly points to the phenyl ring center, forming a hydrogen bond-like conformation (Figure 1d). The conformations were located at the region with *d*_NM_ ~3.3 Å, θ ~10° and ω ~150°, well within the empirical cutoffs for NH–π interactions (*d*_NM_ < 4.3 Å, −25° < θ < 25°, and ω > 120°) extracted from the previous data-mining of protein structures [1].

Except for the bond-conformation parameters, side-chain-position parameters, *d*_Cγ-π_, τ and χ, describe relative positions between the two molecules or side chains (Figure 1d). The separation distances were located at the regions with *d*_Cγ-π_ ~5 Å, τ ~150° and χ ~93°. There was a third pair between Gln in the trailing and Phe in the leading chains. The side chains were pointed in opposite directions (*d*_NM_ = 7.4 Å), which did not form any interactions.

### 3.3. Energetic Favorability of Amide–π Interactions Characterized with Quantum Methanics

To explore whether the Q-F pairs captured in the crystal structures are energetically favored, ab initio QM calculations were performed on an acetamide-toluene complex as a model system of Gln–Phe interactions. A global minimum of interaction energy was obtained at the M06-2X/aug-cc-pVTZ level by scanning the relative positions of the acetamide atom N, representing Gln Nε, with respect to the coordinate set on the phenyl ring (Appendix A). The global minimum of interaction energy (ΔE) was −5.81 kcal/mol similar to hydrogen bonding (approximately −5 kcal/mol) [40,41] and smaller than cation–π interactions (about −19 kcal/mol) [42].

In order to better understand the relationship between the conformation and interaction energy (ΔE), geometric parameters were plotted against ΔE (Figure 2a). The optimal (ΔE = −5.81 kcal/mol) and sub-optimal (−5.75 < ΔE < −5.48 kcal/mol) conformations were located at the region with *d*_NM_ ~3.2 Å, θ ~10° and ω ~150°, were highly similar to those of axial Q-F pairs in the crystal structure, which suggested the latter should also be energetically favored (Table 1). At 0 < θ < 20°, ΔE was plotted with respect to *d*_NM_ and the curve was similar to the Leonard–Jones potential. The minimal ΔE occurs at *d*_NM_ ~3.0–3.5 Å. Beyond *d*_NM_ ~5 Å, there was little interaction, while within *d*_NM_ ~3 Å, ΔE increased dramatically (Figure 2c). This result indicated that the NH–π interaction was favorable in specific geometry of distance and orientation.

The side-chain-position parameters of optimal and sub-optimal states were located at the regions with *d*_Cγ-π_ ~5 Å, τ ~50° and χ ~66°, indicating that acetamide (Gln Cγ-Nε) was tilted against the phenyl ring (Figure 2b). Notably, the sub-optimal states were clustered near the toluene C atom (Phe Cβ) with χ ~65°, which might enhance interaction between acetamide and toluene methyl group. With 40° < τ < 50°, the data of *d*_Cγ-π_ vs. ΔE were projected on a plane (Figure 2d). ΔE decreases as *d*_Cγ-π_ decreases. When 4.5 Å < *d*_Cγ-π_ < 5 Å, ΔE decreases as τ increases (Appendix A). This suggests that, although the side chain proximity and orientation could affect the interaction energy, an NH–π interaction could be formed under various side chain conformations in protein.

### 3.4. Balanced Contributions from Various Types of Chemical Forces

To further understand the chemical property of NH–π interactions, the single point energy of the global minimum of acetamide-toluene was calculated at a higher level of DLNPO-CCSD(T)/aug-cc-pVQZ. The interaction energy was −5.95 kcal/mol. Symmetry-adapted perturbation theory (SAPT) was then used to separately examine the electrostatic (*E*_ele_), exchange (*E*_ee_), induction (*E*_ind_), and dispersion (*E*_disp_) terms that contribute to the Q–F interaction energy (Appendix A). The SAPT2+3/aug-cc-pVQZ calculation revealed a total interaction energy of −7.03 kcal/mol, with 1 kcal/mol overestimated from the CCSD(T) value. Both *_E_*_ele_ (−6.6 kcal/mol) and *E*_disp_ (−8.5 kcal/mol) made comparably large contributions (Appendix A). The *E*_disp_/*E*_ele_ ratio (1.29) indicated a relatively balanced interaction between dispersion and electrostatic energies that indeed affected characteristics of both the protein interior and exterior [43].

### 3.5. Environmental Dielectric Constants

To explore how the NH–π interaction is influenced under a wide range of environments, such as protein interior and aqueous solution, the DLNPO-CCSD(T) calculations with implicit solvent model CPCM were performed. The interaction energy moderately decreased to −4.54 kcal/mol in protein interior (dielectric constant ε = 4) and −4.03 kcal/mol in water (ε = 78.4), respectively, compared to −5.95 kcal/mol in a vacuum. In contrast, cation–π interactions, such as NH4^+^-benzene, were much more favorable in vacuum (approximately −19 kcal/mol) but significantly weakened to −3.6 kcal/mol in protein interior and −1.3 kcal/mol in water, which displays a high dependency on the polarity of the environment [42].

### 3.6. Conformational Distribution Sampled by Molecular Dynamics Simulations

In addition to the energetic favorability revealed by QM, atomistic molecular dynamics (MD) simulations were performed with the crystal structure of the collagen triple helix to investigate the dynamics of NH–π interactions in solution. To this end, the geometric parameters were extracted from the 200 ns MD trajectory and plotted as color-coded density maps (Figure 3). For the two closely contacted Q-F pairs, ~22% of the bond conformations, represented by *d*_NM_, θ, and ω, fell within the empirical cutoffs. Due to the flipping of the phenyl ring, the τ values could be either positive or negative and there were two symmetrical peaks around the line τ=0° on the *d*_Cγ-π-τ_ plane (Figure 3b). The most frequently occurring conformations in the trajectory had the similar geometric values to those in the crystal structure and QM models (Table 1). The MD data suggested that the Q-F pairs in the crystal structure should be maintained in solution to stabilize the protein, consistent with the observations in the CD experiments. For the third pair of Gln in the trailing and Phe in the leading chain, no interaction was detected in the MD trajectories (Appendix A).

### 3.7. Data Mining of Protein Structures

In order to survey the situation of NH–π interactions between Gln and Phe in proteins, a non-redundant protein structure dataset was built consisting of 9658 well-resolved (resolution ≤ 2 Å) crystal structures. Among the 1747 Q-F pairs (*d*_NM_ ≤ 4.5 Å) in the dataset, ~5% of the pairs (94 out of 1747) fell within the cutoffs that defined NH–π interactions (*d*_NM_ < 4.3 Å, −25°< θ < 25°, and ω > 120°). The phenyl rings of the 94 Q-F pairs were superimposed and formed two umbrella-shaped distributions of Gln above and beneath the ring, where all of the Nε-H groups were positioned approximately perpendicular to the ring center with similar *d*_NM_, θ and ω values (Figure 4). The values of *d*_Cγ-π_ and τ range from 2.9 Å to 5.9 Å and −74.7° to 56.2°, respectively. The Gln side chains in the axial Q-F pairs were located in the center of the umbrella and had extremely high *d*_Cγ-π_ (~5.3 Å) and τ (~61.1°). Notably, rather than uniformly distributed around the phenyl ring, the Gln side chains are less frequently projected toward the Phe Cα with χ ranging from 4.6° to 175.7°, potentially preventing steric clashes. This finding suggests that NH–π interactions could occur in proteins through diverse side chain conformations within a wide range of distances between Gln and Phe.

It is noted that Phe is hydrophobic while Gln is hydrophilic, which is relevant to their functions in the protein. To explore locations of the NH–π interactions between them in proteins, i.e., the hydrophobic cores or the protein surfaces, the solvent accessible surface (SAS) area of the Q-F pairs were calculated with DSSP [44,45]. The majority (56 out of 94) of Q-F pairs were buried relatively deep inside the protein core (SAS < 30 Å^2^) (Figure 4d). Eighteen Q-F pairs were partially exposed to solvent (60 Å^2^ < SAS < 120 Å^2^). Gln had higher SAS than Phe (Figure 4e). Except for the Q-F pairs in peptide GPQGFO, there were three fully exposed pairs (Appendix A). In one case, Gln and Phe were located in the i and i+4 positions of α-helices, which may stabilize the α-helices. SAS analysis indicated that the Gln-Phe NH–π interactions could occur in both the core and the surface of proteins. Consistently, when the interaction was transferred from a vacuum to either low or high dielectric constant, simulating the protein core or surface, energy losses of only ~33% and ~25% were detected, respectively.

## 4. Conclusions

From thermal unfolding experiments, we have demonstrated that NH–π interactions allow the side chain of Gln to interact with the π-system of Phe in an axial geometry that is less influenced by the environment. The interaction is energetically highly favorable with its magnitude similar to hydrogen bonding [40,41]. Computational and experimental data from structural biology, QM, and structural bioinformatics demonstrate that the polar Nε-H group is typically positioned perpendicular to the phenyl ring and aligned with the phenyl ring center, which is well consistent with a previous study [1].

Two major features of the NH–π interactions have been revealed. First, the NH–π interactions are stable in various environments, i.e., the interaction energy is insensitive to dielectric constants. The stabilizing effect also remains almost constant under a wide range of pH and ionic strengths. When transferred from a vacuum to either low or high dielectric constant, the NH–π interaction loses only ~33% and ~25% energy, respectively, while cation–π interaction decays by ~70% and ~90% [42]. The Q-F pairs, both buried in the hydrophobic core or exposed on the protein surface, are identified in the data mining of the protein structures.

Secondly, the NH–π interactions can be formed by diverse side chain conformations under a range of distances between Gln and Phe. In the GPQGFO structure, Gln and Phe, with Cα atoms separated by 7.7 Å, made a strong NH–π interaction. Multiple sub-optimal interacting conformations were observed in the QM models.

Similar interactions between side chain amino and aromatic groups, such as between Asn and Phe, can be further explored, which will enable the use of NH–π interactions in the design of hydrophobic protein cores, solvent accessible protein surfaces, and protein–protein binding interfaces.

## Figures and Tables

**Figure 1 biomolecules-12-01433-f001:**
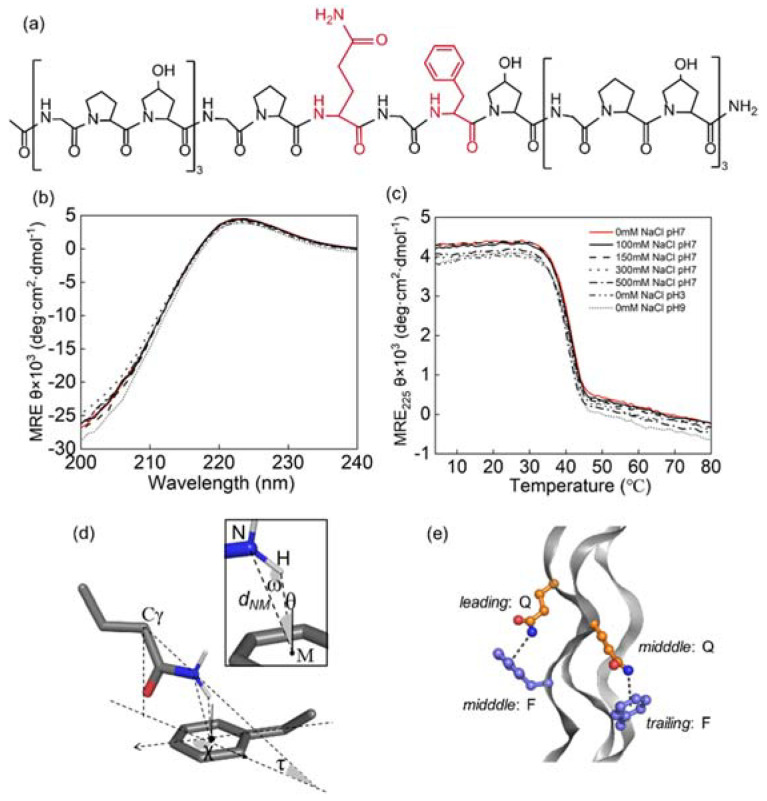
Experimental characterization and geometric description of Gln–Phe interactions: (**a**) primary sequence of the collagen-like peptide, [(GPO)_3_GPQGFO(GPO)_3_]_3_ is shown, where O represents (4R)-hydroxyproline. Gln and Phe are colored in red; (**b,c**) triple helix formation and stability of peptide GPQGFO were characterized by CD in 10 mM phosphate buffer under various NaCl concentrations and pH values as labeled in the plot. Wavelength spectra at 4 °C are shown in (**d**), and temperature melting experiments monitored at 225 nm from 4 to 80 °C are shown in (**e**); (**d**) geometric parameters of an NH–π interaction in either a Q-F pair or an acetamide-toluene complex are defined. Three side-chain-position parameters, *d*_Cγ-π_, τ, and χ, describe relative position and orientation between Gln Cγ-Nε (acetamide C-N) and the phenyl plane. Inset on top-right corner shows three bond conformation parameters, *d*_NM_, θ, and ω, describing how Gln Nε-H (acetamide N-H) is positioned relative to the phenyl ring center M; (**e**) the crystal structure of the collagen triple helices was solved (PDB ID:7VEG). Two closely contacted Q-F pairs are shown in ball-and-stick representation. The chain identities are as labeled in figure (**d**,**e**), figures were generated with Pymol.

**Figure 2 biomolecules-12-01433-f002:**
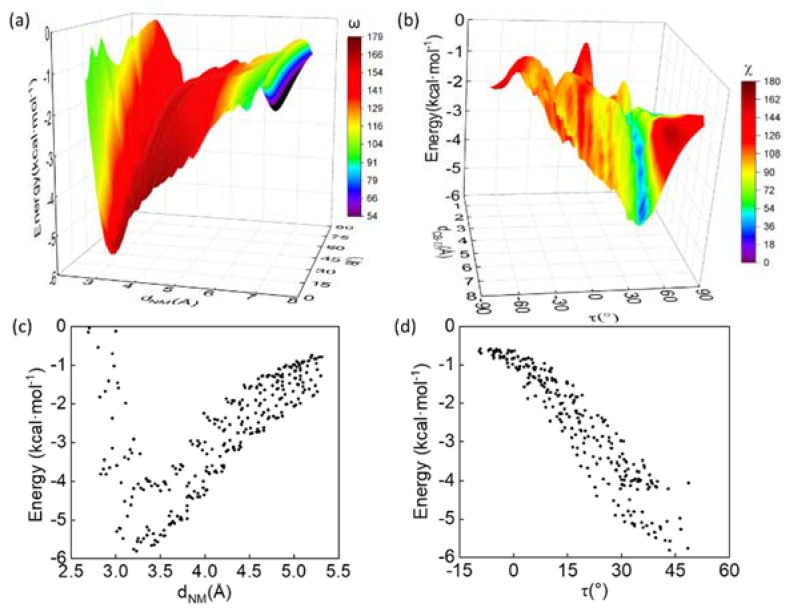
(**a**) Three-dimensional (3D) interaction energy (ΔE) surface is plotted against *d*_NM_ and θ. The energy surface is color-coded with ω. The color scales are located on the right. (**b**) 3D ΔE surface is plotted against *d*_Cγ-π_ and τ. The energy surface is color-coded with χ. The color scales are located on the right. (**c**) The data points with 0 < θ < 20° in the 3D-plot in (**a**) are projected on the ΔE-*d*_NM_ plane. (**d**) The data points with 40° < χ < 50° in the 3D-plot in (**b**) are projected on the ΔE-τ plane.

**Figure 3 biomolecules-12-01433-f003:**
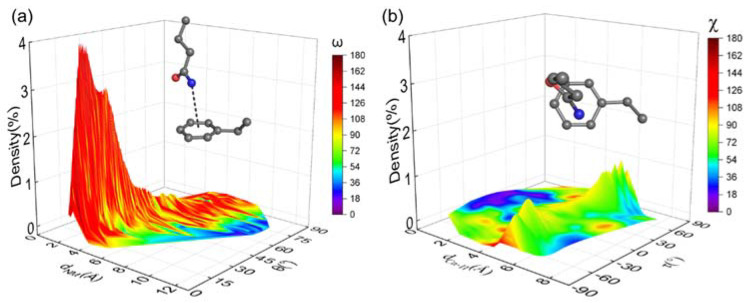
Molecular dynamics simulation of the collagen-like peptide GPQGFO. The geometric parameters of the Q-F pair between the leading and middle chain were extracted from the trajectory and plotted into three-dimensional density maps. (**a**) Conformation occurring percentage is plotted against *d*_NM_ and θ. The surface is color-coded with ω. (**b**) Conformation occurring percentage is plotted against *d*_Cγ-π_ and τ. The surface is color-coded with χ. The color scales are located on the right. Insets of (**a,b**) show the most occurring conformations.

**Figure 4 biomolecules-12-01433-f004:**
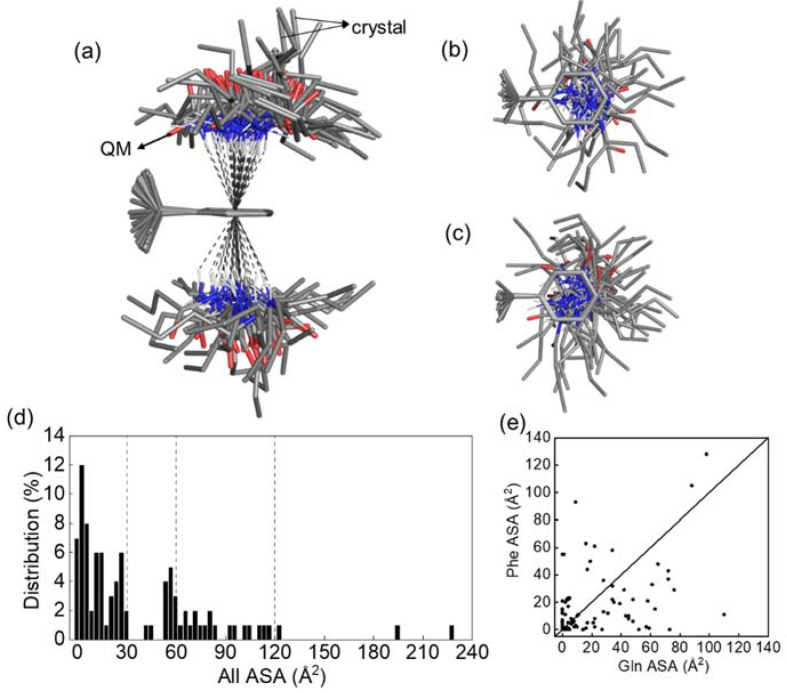
Q-F pairs with a typical NH–π interaction extracted from the non-redundant dataset of protein crystal structures. (**a**–**c**) The Q-F pairs are positioned by superimposing Phe with each other. The side, bottom-up, and top-down views are shown in (**a**), (**b**) and (**c**), respectively. (**d,e**). Accessible surface (ASA) of the Q-F pairs were calculated. Distribution of the total ASA is shown in (**d**). The scatter plot of ASA of Gln vs. Phe is shown in (**e**).

**Table 1 biomolecules-12-01433-t001:** Geometric parameters of Q-F pairs extracted from QM calculation of the acetamide-toluene complex, crystal structure and MD trajectories of peptide GPQGFO, and a non-redundant dataset of protein crystal structures.

Methods	Pairs	*d*_NM_ (Å)	θ (°)	ω (°)	*d*_Cγ-Π_ (Å)	τ (°)	χ(°)
QM	Acetamide-Methylbenzene	3.2	8.2	149.1	4.9	46.7	65.9
Crystal	*L*^a^:Q-*M*^a^:F	3.3	8.4	159.2	5.3	61.1	93.0
	*M*^a^:Q-*T*^a^:F	3.5	20.9	141.9	5.5	62.3	107.8
MD ^b^	*L*^a^:Q-*M*^a^:F	3.3	15.9	145.4	5.6	72.3	43.8
					5.2	−50.5	115.1
	*M*^a^:Q-*T*^a^:F	3.2	10.5	139.9	5.5	73.0	113.4
					5.5	−78.4	122.4
Data mining ^c^	Q-F	3.5 ± 0.23	14.1 ± 6.1	146.3 ± 14.1	3.9 ± 0.8	72.1 ± 19.8	103.9 ± 37.1
					3.9 ± 0.6	−16.0 ± 14.9	122.2 ± 43.5

^a^ Leading, middle and trailing chain of the triple helices were represented as L, M, and T, respectively. ^b^ The parameter values with the highest occurring frequencies were listed here. ^c^ The mean and standard deviation values were listed here.

## Data Availability

Not applicable.

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
