# Peer review of "Structural Achievability of an NH–π Interaction between Gln and Phe in a Crystal Structure of a Collagen-like Peptide"

_biomolecules, 2022, doi:10.3390/biom12101433_

Round 1

Reviewer 1 Report

The manuscript from Zhang et al titled “Structural achievability of an NH-π interaction between Gln and Phe in a crystal structure of a collagen-like peptide” investigates the effect of the interaction of uncharged amino groups with conjugated π -systems in collagen structures in particular and protein structure in general. The authors incorporated an NH-π interaction between a phenylalanine and glutamine residue into a well-known collagen model peptide system and tested its influence on stability by melting curve measured by CD. They also determined the crystal structure of the investigated collagen model peptide and used this to better explain their observations. Additionaly, they included quantum mechanical calculation on acetamide-toluene complexes as a model for their observed Q-F interaction and molecular dynamics calculation on both observed Q-F interaction from their crystal structure. To apply their findings to non-collagen proteins structures they performed a database search for crystals structure bearing these amino acids interactions and report on the general configuration and placement in the overall protein structure.

Overall, the authors conclude that the NH-π interaction has a stabilising contribution to protein stability with an average contribution of common H-bridges (~5 kcal/mol). They also claim that – in contrast to ionic interactions - NH-π interactions are nearly independent of the polarity of their respective surroundings and span a wide area of distances. In total they conclude the NH-π interactions might be important in protein stability and should be considered in protein design.

The manuscript is fairly written; however some smaller correction should be made:

* Page 4 Line190f: Sentence is not complete.

* It is much more common to refer to literature, by their first author (not the last as the authors do) and I would recommend changing this.

There also some minor typing errors / mistakes which should be corrected:

* Table S2: Typo in Refinement/ Resolution

* Page5 L217: refers to a wireframe model available in the SI, however, this is not present.

* Fig 1b has MRE225 at the X-Axis, this should surely be MRE.

* Page2 L93: Do not use underscore as this masks the difference between Q and O, use bold (or italics) only.

* Page2 L91: The authors refer to sequences of peptides, however only one peptide is made.

The crystal structure cannot be completely evaluated at this moment, as the authors did not submit any validation report. So NO final decision on the structure can be performed currently!

However, the formal resolution of 1.4 with a huge signal (I/sigI > 5) is indicative for very nice data. In the text the authors even state diffraction up to 1.25 A, why is only 1.4 used in refinement?

The authors used 5NAY for molecular replacement – a literature reference should be included as well.

From what can be said from the little information available, the overall structural solution seems fine. Please also include references for the used software suites (COOT, Phenix).

Which software was used for generating the figures (+ reference)?

Furthermore, please add the following info to the Table S2:

* CC½    

* Highest resolution shell (in parenthesis behind refinement resolution)

* Number of reflections in the test set (in parenthesis behind No. reflections)

* Correct the Spacegroup to P212121

* Mention the PDB deposition code (7VGE?) in the M&M sections

The value of 20% of Rfree at this high resolution seems to be rather on the high end. Especially with the 4% difference to RWork. Can the authors comment, how they explain these values? Did the authors try anisotropic refinement of the protein side chains?

As the orientation of the individual side chains in Fig 1e is very important a figure showing the density of this configuration should be added to the SI.

In conclusion

This manuscript references highly to Walker et al 2021 and while this is mentioned and references occasionally some references are missing (e.g. P4L190, P5L218). The manuscript would also benefit of more refined statement on the scientific gap (P2L74-79). This paragraph should state more precisely what exactly was not covered in the above mentioned paper and what this manuscript tries to show.

In general, I think this manuscript adds some interesting data to the field of structural biology. It extends the fundamental work by the research group of Prof. Hartgerink by some experimental data, which were missing earlier.

Reviewer 2 Report

The manuscript titled "Structural achievability of an NH- interaction between Gln 2 and Phe in a crystal structure of a collagen-like peptide" by Zhuang et al., has demonstrated the role of NH-pi interactions in the collagen like peptide. The work is well-designed and experiments are systematically carried out. Except few corrections, the manuscript can be accepted for publication.

Comments are as follows:

1. In page 2, line 90, reflection electron microscope is repeated twice. 

2. In Circular dichroism studies, it is mentioned that the spectra were recorded between 190-260 nm but in the figure 1b, it is given from 215 to 240 nm. It would be better to give from 190 to 240 or 260 nm so that it will be easy to know that the triple helical structure of collagen-like peptide is maintained

3. In Fig S5, in the mass spectral data, the other peaks need to be assigned.

4. It will be beneficial for the readers if the authors do the peptide sequence analysis for the chosen collagen-like peptide.

Round 2

Reviewer 1 Report

The authors have significantly improved their manuscript and dealt with most of my suggestions. The manuscript was in principle sound during initial submission, but lacked attention to details and had a couple of inconsistencies. Unfortunately, this problem still exists and with the now available PDB validation report, even a couple of new ones had surfaced. I can only once more suggest, that the authors carefully go through the manuscript and iron out all the obvious inconsistencies which are most likely caused by rushed writing.

The validation report only accounts for 483 atoms in the protein and 115 water atoms. In the table S2 988 are stated. If the latter number includes hydrogens (very unusual, especially at this resolution!) this should be stated (or the number corrected).

The validation report states P21221 as the space group. In the last revision the authors had P212121 in the manuscript and P21221 in the Table S2. Care should be taken to report the correct space group!

The newly added “highest resolution shell” shell, should reach out to the highest reported resolution (1.38Å). It makes mathematically no sense that it ends and 1.40Å.

The completeness reported in the validation reports is 90.7 overall (as submitted by the authors) but reported 98.1 in Table S2.

The I/sigI at 46 seems rather high and is only 4.6 reported by the PDB validation reports (although that is error prone to the submission of reduced structure factors). Is this number really the correct one for the WHOLE dataset or maybe just for the lowest resolution shell?

The validation report states 970 reflections in the Rfree test set (a number I asked for in my first review). This is a very solid basis for statistics, however, it reflects 10% of the initial data. This should be changed in the legend of the Table S2.

In the validation report the Rfree is calculated to be only 18%, 2 percent points lower than reported by the authors in table S2. If this is not due to the mix-up of RFree sets during submission (this would also impact above mentioned point), this would make a much better statistic. The authors should re-evaluate why the PDB validation report pipeline receives better statistics.
